

# PER2 regulates odontoblastic differentiation of dental papilla cells *in vitro via* intracellular ATP content and reactive oxygen species levels

Haozhen Ma, Xinyue Sheng, Wanting Chen, Hongwen He, Jiawei Liu, Yifan He and Fang Huang

[1] Hospital of Stomatology, Sun Yat-sen University, Guangzhou, China
[2] Guangdong Provincial Key Laboratory of Stomatology, Guangzhou, China
[3] Guanghua School of Stomatology, Sun Yat-sen University, Guangzhou, China

## ABSTRACT

**Background**. Dental papilla cells (DPCs) are one of the key stem cells for tooth development, eventually forming dentin and pulp. Previous studies have reported that PER2 is expressed in a 24-hour oscillatory pattern in DPCs *in vitro*. *In vivo*, PER2 is highly expressed in odontoblasts (which are differentiated from DPCs). However, whether PER2 modulates the odontogenic differentiation of DPCs is uncertain. This research was to identify the function of PER2 in the odontogenic differentiation of DPCs and preliminarily explore its mechanisms.

**Methods**. We monitored the expression of PER2 in DPCs differentiated *in vivo*. We used PER2 overexpression and knockdown studies to assess the role of PER2 in DPC differentiation and performed intracellular ATP content and reactive oxygen species (ROS) assays to further investigate the mechanism.

**Results**. PER2 expression was considerably elevated throughout the odontoblastic differentiation of DPCs *in vivo*. Overexpressing *Per2* boosted levels of odontogenic differentiation markers, such as dentin sialophosphoprotein (*Dspp*), dentin matrix protein 1 (*Dmp1*), and alkaline phosphatase (*Alp*), and enhanced mineralized nodule formation in DPCs. Conversely, the downregulation of *Per2* inhibited the differentiation of DPCs. Additionally, downregulating *Per2* further affected intracellular ATP content and ROS levels during DPC differentiation.

**Conclusion**. Overall, we demonstrated that PER2 positively regulates the odontogenic differentiation of DPCs, and the mechanism may be related to mitochondrial function as shown by intracellular ATP content and ROS levels.

Corresponding authors
Yifan He, 731264038@qq.com,
heyf59@mail.sysu.edu.cn
Fang Huang, hfang@mail.sysu.edu.cn

## INTRODUCTION

Tooth development is a complicated process involving both the epithelium and mesenchyme, as well as the precise temporal and spatial roles of molecular signaling (*Sui et al., 2023*; *Yu & Klein, 2020*; *Yuan & Chai, 2019*). Dental papilla cells (DPCs), a type

of mesenchymal stem cells (MSCs) of cranial neural crest origin, are located beneath the enamel organ and progressively wrapped by the enamel organ as tooth germs develop (*Kim et al., 2012*; *Tziafas & Kodonas, 2010*). Induced by the inner enamel epithelium and promoted by molecular signals, outer DPCs can further differentiate into odontoblasts that synthesize and secrete dentin, whereas inner DPCs are undifferentiated and surrounded by dentin, becoming pulp cells (*Rothová, Peterková & Tucker, 2012*; *Yu & Klein, 2020*; *Yuan & Chai, 2019*). DPCs eventually form pulp and dentin, which are the main components of teeth. Therefore, the differentiation of DPCs is crucial for tooth development and dentin formation. However, the molecular signaling mechanisms by which DPCs differentiate into odontoblasts remain unclear.

Circadian rhythms are important environmental factors involved in tooth development. Tooth development follows circadian rhythms (*Ohtsuka & Shinoda, 1995*), with periodic growth lines in both the enamel and dentin (Retzius's lines & Von Ebner's lines) (*Molnar & Ward, 1975*). Animal experiments have shown that the disruption of circadian rhythms can seriously compromise tooth development (*Huang et al., 2021*; *Ohtsuka-Isoya, Hayashi & Shinoda, 2001*). Major clock genes, *Bmal1* and *Per2*, were oscillating in a regular 24-hour circadian rhythm in tooth-derived stem cells, suggesting that *Bmal1* and *Per2* may be regulators of tooth development (*Furukawa et al., 2005*; *Hilbert et al., 2019*; *Jiang et al., 2022*; *Lacruz et al., 2012*; *Zheng et al., 2013*). The role of *Bmal1* in promoting oral and maxillofacial development has been previously investigated (*Hirai et al., 2018*; *Koshi et al., 2020*; *Liu et al., 2022*; *Zhao et al., 2018*). In particular, *Bmal1* was recently found to regulate dental pulp cell differentiation and dentin formation (*Xu et al., 2022*). However, PER2 has rarely been studied in dentin formation and tooth development. PER2 regulates enamel development (*Huang et al., 2021*). In addition, strong PER2 expression in odontoblasts (which are differentiated from DPCs) has been observed *in vivo* at different time points of tooth development (*Zheng et al., 2011*). Additionally, oscillatory expression of PER2 over a 24-hour period has been observed in undifferentiated and differentiated DPCs *in vitro* (*Jiang et al., 2022*). On the basis of the aforementioned research, we speculate that PER2 is involved in the modulation of DPC differentiation and dentin formation, but its specific role and underlying mechanism remain unclear.

Mitochondria are functional organelles that provide energy to cells. Enhanced mitochondrial function and an energy metabolism paradigm shift (glycolysis-oxidative phosphorylation) are key for MSC differentiation (*Ji et al., 2020*; *Yan, Diao & Fan, 2021*). We revealed in preliminary research that DPC differentiation was associated with enhanced mitochondrial function, as evidenced by increased mitochondrial membrane potential (MMP), elevated intracellular ATP content, and decreased reactive oxygen species (ROS) (*Zhang et al., 2018*). Furthermore, mitochondria-related molecules, such as malic enzyme 2 (a mitochondrial NAD-dependent enzyme) (*Zhang et al., 2023*), optic atrophy 1 (OPA1, a mitochondrial fusion protein) (*Zhang et al., 2023*), and SIRT4 (a mitochondrial sirtuin) (*Chen et al., 2021*), were also found to modulate DPC differentiation. Inhibition of mitochondrial respiratory function hinders DPC differentiation (*Zhang et al., 2018*). The aforementioned results suggest that the differentiation of DPCs is closely linked to mitochondrial function. PER2 has been implicated in physio/pathological pathways in

erythrocytes (*Sun et al., 2017*), colonic epithelial cells (*Chen et al., 2022*), cardiomyocytes (*Weng et al., 2021*), and hepatocytes (*Chen et al., 2009*) *via* the regulation of mitochondrial function, according to recent studies. However, how PER2 affects mitochondrial function in the process of DPC differentiation has little been reported.

This research aimed to identify the function of PER2 in odontogenic differentiation of DPCs and further reveal the impact of PER2 on mitochondrial function during DPC differentiation. First, PER2 expression was assessed as DPCs differentiated *in vivo*. Next, we assessed the function of PER2 in odontogenic differentiation of DPCs by overexpressing and knocking down *Per2 in vitro*. Finally, intracellular ATP content and ROS levels were assessed to preliminarily examine the potential mechanisms through which PER2 modulates DPC differentiation.

## MATERIALS & METHODS

### Immunohistochemistry

The present study was approved by the Institutional Animal Care and Use Committee, Sun Yat-sen University, China (No. SYSU-IACUC-2020-000511) and the Ethical Review Committee, Hospital of Stomatology, Sun Yat-Sen University, China (ERC-2013-15). This study adhered to the Ethical Principles of Animal Experimentation.

Normal-growth Sprague-Dawley (SD) rats, 3–5 days postnatal, provided by Sun Yat-sen University, Guangzhou, China, and routinely housed without experimental intervention, were euthanized by neck dislocation after $CO_2$ over-inhalation. Their mandibles were isolated. Fix mandibles in 4% paraformaldehyde, followed by decalcification with 0.5 M EDTA. Tissue was dehydrated, paraffin-embedded, and serially sectioned at 4 $\mu$m. Next, sections were deparaffinized and rehydrated. The slices were immersed in Tris-EDTA heated at 70 °C (5 min), cooled (5 min), heated at 40 °C (10 min), followed by cooling and then incubation in 0.4% pepsin at 37 °C (30 min) for antigen retrieval. After incubating in 3% hydrogen peroxide solution (25 min protected from light), block the slices in 3% bovine serum albumin (BSA). Next, slices were incubated with anti-PER2 antibody (1:200) overnight at 4 °C, as well as in horseradish peroxidase-conjugated (HRP-conjugated) Goat Anti-Rabbit IgG (H+L) for 50 min. Sections were visualized *via* diaminobenzidine in darkness and counterstained with hematoxylin. Sections were visualized by an Aperio AT2 digital pathology scanner (Leica, Weztlar, Germany). Semi-quantitative analysis was performed using the ImageJ 1.51k (NIH, MD, USA) to assess mean optical density. Reagents not specified in this section were purchased from Servicebio, Wuhan, China.

### Cell culture and differentiation

Postnatal 1–2-day SD rats from the above sources were euthanized using the method described above. Isolate their dental papilla tissues under stereomicroscope, as described previously (*Zhang et al., 2018*). Dental papilla tissues were microscopically cut and digested with 3 mg/mL of collagenase type I + 4 mg/mL of Dispase® II (Sigma-Aldrich, St. Louis, MO, USA) for 15 min to produce tissue fragments and cell suspensions of primary cells, which were inoculated in 10-cm culture dishes maintained in $\alpha$-minimal essential medium ($\alpha$-MEM) comprising 20% fetal bovine serum (FBS), 100 U/mL penicillin, and 100 $\mu$g/mL

streptomycin at 37 °C in 5% CO2 humidified air. Passage DPCs by TrypLE when they reached 90% confluence and purified by the difference digestion method. After passaging, DPCs were maintained in basal medium containing 10% FBS. Passages 2-4 were applied to the subsequent experiments. FBS, $\alpha$-MEM, penicillin, streptomycin and TrypLE were products of Gibco, Grand Island, NY, USA.

For differentiation of DPCs, DPCs were grown in osteo/odontogenic induction medium (OS; $\alpha$-MEM supplemented with 10% FBS, 10 mM $\beta$-glycerophosphate, 0.2 mM ascorbic acid and 0.1 mM dexamethasone; Sigma-Aldrich, St. Louis, MO, USA).

For all subsequent cell experiments, sample collection and detection were performed at the same time points by the same personnel.

## Immunofluorescence

DPCs were seeded in 35 mm glass-bottom culture dishes. Fix DPCs with 4% paraformaldehyde for 15 min, and permeabilized using 0.5% Triton X-100 for 15 min. 5% BSA was applied to block DPCs. Cells were incubated with anti-cytokeratin antibody (1:100) or anti-vimentin antibody (1:100) or anti-PER2 antibody (1:100) overnight at 4 °C. Then, DyLight 488 or DyLight 594 conjugated secondary antibody was used to incubate cells for 30 min without light. If cell morphology was required, incubate DPCs with Actin-Tracker Green (Beyotime, Shanghai, China). Finally, DAPI was used to incubate cells for 5 min protected from light to stain cell nuclei. Cells were preserved in PBS and visualized by confocal laser scanning microscope (FV3000; Olympus, Tokyo, Japan).

## Flow cytometry

Digest DPCs using TrypLE and prepare the single cell suspension with PBS comprising 2% FBS. DPC single cell suspensions were incubated with flow cytometry CD45, CD34, CD44, CD90, CD29 antibodies and their isotype controls for 30 min at 4 °C (Elabscience, Wuhan, China). LSRFortessa flow cytometer (BD Biosciences, Franklin Lakes, NJ, USA) was for measurement of fluorescence intensity of cell samples.

## Cells transfection

To overexpress *Per2*, DPCs were seeded at $1.2 \times 10^5$ cells/well in 12-well plates and maintained in basal medium without antibiotics. Within 24 h, cells reached about 80% confluence for transfection. The transfection protocol for each well was as follows: Opti-MEM™ (62.5 μL) diluted with 1.25 μg of plasmid and 2.5 μL of P3000™ reagent was used. Plasmids included pcDNA3.1-PER2 plasmid or negative control pcDNA3.1-NC plasmid. Diluted DNA was added 1:1 to each tube of diluted Lipofectamine™ 3000 and incubated for 10–15 min. The DNA-lipid complex was added to DPCs. The medium (without antibiotics) was replaced after 4–6 h. The culture medium was changed to conditioned medium (OS or basal medium) without antibiotics after 24 h.

To knock down *Per2*, DPCs were seeded at $7 \times 10^4$/well in 12-well plates. Small interfering RNAs (siRNAs) targeting *Rattus Norvegicus Per2* were transfected into cells using a protocol similar to the overexpression protocol, but without the addition of P3000™ reagent. Final siRNA concentration was 50 nM. Opti-MEM™, P3000™ and

**Table 1  RT-qPCR primers.**

| Gene | Species | Forward primer (5′–3′) | Reverse primer (3′–5′) |
| --- | --- | --- | --- |
| PER2 | Rat | AGGGCATTACCTCCGAGTATATC | AGGCGTCCTTCTTACAGTGA |
| Dspp | Rat | ACAGCGACAGCGACGATTC | CCTCCTACGGCTATCGACTC |
| Dmp1 | Rat | CTGGTATCAGGTCGGAAGAATC | CTCTCATTAGACTCGCTGTCAC |
| Alp | Rat | GGAAGGAGGCAGGATTGA | TCAGCAGTAACCACAGTCA |
| Gadph | Rat | TATGACTCTACCCACGGCAAGT | ATACTCAGCACCAGCATCACC |

Lipofectamine™ 3000 were supplied by Invitrogen (Carlsbad, CA, USA). Plasmids and siRNAs were provided by MHBIO (Guangzhou, China).

## RT-qPCR

Real-time quantitative polymerase chain reaction (RT-qPCR) was used for assessment of DPC gene expression. Total RNA was extracted *via* an RNA extraction kit from Yishan Biotechnology, Shanghai, China. Qualified RNA samples were reverse transcribed into cDNA. After proportionate addition of SYBR Green, primers (listed in Table 1), and RNA samples, RT-qPCR was conducted by Roche LightCycler® 96 System. The reverse transcription reagents and SYBR Green were purchased from Yeasen (Shanghai, China). The amplification program was set up according to the procedures recommended in the instructions. Glyceraldehyde-3-phosphate dehydrogenase (GADPH) acted as an internal reference for normalization. Calculate relative expression of target genes according to the $2^{-\Delta\Delta Ct}$ method.

## Western blotting

Total protein was extracted from DPCs using radioimmunoprecipitation assay (RIPA) lysis buffer containing phenylmethylsulfonyl fluoride (PMSF). The extracted proteins were quantified *via* a bicinchoninic acid (BCA) protein assay kit. RIPA, PMSF and BCA kit were purchased from Cwbio, Beijing, China. Equal amounts of protein lysates were dispersed by SDS-PAGE and transferred to PVDF membranes. The membranes were blocked in skim milk and incubated overnight at 4 °C with the following antibodies: anti-PER2 antibody (1:2000), anti-DSPP antibody (1:500), anti-DMP1 antibody (1:1000), and anti-$\beta$-actin (1:1000). Next, the membranes were incubated in the HRP-conjugated secondary antibodies corresponding to the primary antibody species (1:2000) for 45 min. Immunoreactive bands were detected using enhanced chemiluminescence (ECL; Millipore, Billerica, MA, USA), captured on the Bio Rad ChemiDoc chemiluminescence imaging system, and quantified using ImageJ 1.51k (NIH, MD, USA).

## Quantitative alizarin red staining

For fixation, cells were treated with 4% paraformaldehyde for 30 min. Next, use Alizarin Red S Solution (ALIR-10001, Cyagen Biosciences, Santa Clara, CA, USA) to stain cells for 5 min, followed by rinsing in double distilled water. An Epson professional-grade scanner was used to capture the stained overview and an Olympus IX83 inverted microscope was applied for capturing mineralized nodules. For semi-quantification of mineralized nodules, 0.1 M hexadecylpyridinium chloride monohydrate was supplemented and shaken for 30

min, and the supernatant was collected to determine the absorbance at 562 nm with a Biotek Epoch 2 Microplate Spectrophotometer.

### ROS assay

After preparation into single cell suspensions, DPCs were treated with CellROX® Green Reagent (Invitrogen, Carlsbad, CA, USA) for 1 h at 37 °C, followed by rinsing in PBS. A BD LSRFortessa flow cytometer was applied for measurement of fluorescence intensity of cell samples.

### Intracellular ATP content assay

Use an ATP assay kit (S0026; Beyotime, Shanghai, China) for measurement of intracellular ATP content. Lysed cell samples or standards were mixed with ATP assay reagents and added to an all-black 96-well plate for chemiluminescence detection by BioTek Synergy H1 multimode microplate reader.

### Statistical analysis

Statistical analyses were undertaken with GraphPad Prism 8.0.2 (GraphPad, San Diego, CA, USA). All experimental data are provided as mean ± standard deviation (SD), derived from at least three independent experiments. A two-tailed Student's $t$-test was applied for two-group comparisons. Multiple group comparisons were assessed by one-way analysis of variance. Fisher's least significant difference test was used for post-hoc analysis. $P < 0.05$ was considered statistically significant.

## RESULTS

### PER2 elevated during odontoblastic differentiation of rat DPCs *in vivo*

Immunohistochemistry was carried out on the incisors of Sprague-Dawley rats on postnatal day 5 (Fig. 1A) to assess whether PER2 has the potential to modulate DPC differentiation and dentin development *in vivo*. As shown in Fig. 1B, a vertical section of the incisor, DPCs close to the enamel organ gradually elongated, polarized and differentiated into odontoblasts. The differentiation of DPCs can be divided into four stages, and the typical morphologies of the cells at each stage are shown in Figs. 1C–1F. In the undifferentiated DPCs (Fig. 1C), little PER2 expression can be observed. As DPCs gradually differentiated into preodontoblasts (Fig. 1D) and immature odontoblasts (Fig. 1E), PER2 expression levels gradually increased. By the time the DPCs had fully differentiated into mature odontoblasts (Fig. 1F), the expression of PER2 was at its highest level. Semi-quantitative analysis revealed that PER2 expression was progressively upregulated as DPCs differentiated *in vivo*. Therefore, it is highly probable that PER2 modulates DPC differentiation.

### Expression and intracellular localization of PER2 in DPCs *in vitro*

After identifying the potential role of PER2 in DPC differentiation *in vivo*, we performed *in vitro* experiments to further assess the regulatory functions of PER2 in DPC differentiation. DPCs were extracted and cultured *in vitro* according to previously described protocols (*Zhang et al., 2018*). After 3 days, primary DPCs migrated from pieces of dental papilla tissue (Fig. 2A) and were successfully amplified. DPCs were homogeneous, large

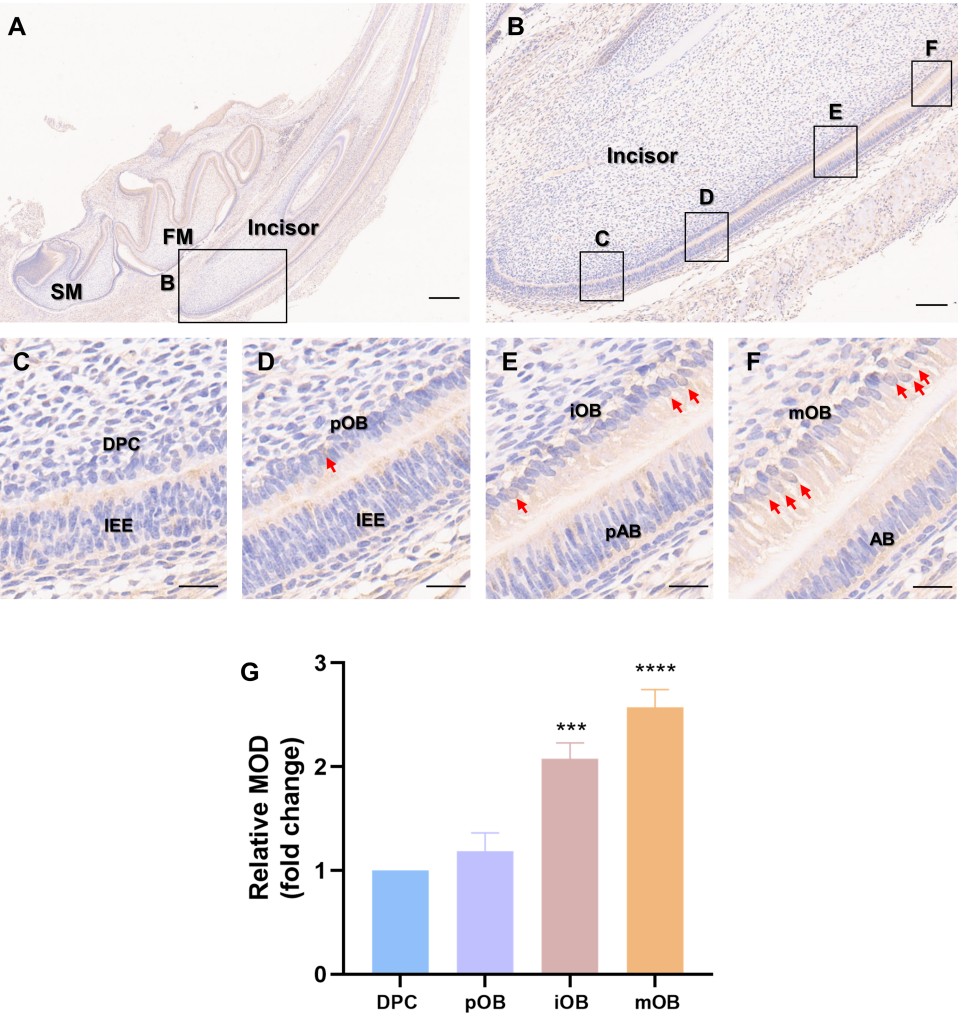

**Figure 1** Patterns of PER2 expression during odontoblastic differentiation *in vivo*. (A) Immunohisto-chemistry of mandible of Sprague-Dawley rats on postnatal day 5. FM, tooth germ of the lower first molar; SM, tooth germ of the lower second molar. Scale bar: 500 μm; (B) Immunochemistry illustrates the expression patterns of PER2 throughout DPCs differentiation. Scale bar: 100 μm; (C–F) the outer cells of dental papilla gradually differentiate, elongate, and polarize, undergoing four stages of differentiation: (C) DPCs, (D) preodontoblasts, (E) immature odontoblasts, and (F) mature odontoblasts; Scale bar: 20 μm. As differentiation proceeds, PER2 expression levels in the DPC-odontoblasts axis are continuously upregulated. Brown staining by HRP-DAB shows the expression of PER2. Red arrows point to cells with strong nuclear localization of PER2. IEE, inner enamel epithelium; pAB, preameloblasts; AB, ameloblasts; DPC, dental papilla cells; pOB, preodontoblasts; iOB, immature odontoblasts; mOB, mature odontoblasts. (G) Mean optical density (MOD) of the DPC, pOB, iOB, and mOB groups. Compared with DPC group, * $p < 0.05$, ** $p < 0.01$, *** $p < 0.001$, **** $p < 0.0001$.

polygonal fibroblast-like cells with abundant cytoplasm and centered nuclei (Fig. 2B). Immunofluorescence illustrated that the DPCs were strongly expressed vimentin (MSCs marker, Fig. 2C) but hardly expressed cytokeratin (epithelial cells marker, Fig. 2D). Flow cytometry showed DPCs positively expressing mesenchymal markers CD44 (92.1%),
CD90 (94.8%), and CD29 (92.9%), but negatively expressing hematopoietic markers CD45 (3.15%) and CD34 (3.62%) (Figs. 2E–2F).

To determine the levels and intracellular localization of PER2 in DPCs *in vitro*, cellular immunofluorescence assays were performed (Fig. 3). Nuclei were stained by DAPI (Fig. 3A), while the outlines of the cells were stained by Actin-Tracker (Fig. 3B). PER2 (red fluorescence) was strongly expressed in the nucleus and perinuclear cytoplasm (Figs. 3C–3D).

## Overexpression of *Per2* enhances odontoblastic differentiation of DPCs *in vitro*

*In vitro, Per2* was overexpressed in DPCs and relevant differentiation markers were examined to identify the regulatory role of PER2 in odontogenic differentiation of DPCs (Fig. 4). PcDNA3.1-PER2 (oe-PER2) was transfected to overexpress *Per2*, as well as pcDNA3.1-NC (oe-NC) for negative control. Overexpression efficiency of *Per2* was measured *via* RT-qPCR and western blotting at 48 and 72-hours post-transfection, respectively (Fig. 4A). MRNA and protein levels of PER2 remarkably elevated in oe-PER2 in comparison to oe-NC, confirming the success of *Per2* overexpression in DPCs (Fig. 4A).

After *Per2* was successfully overexpressed, the levels of indicators associated with odontogenic differentiation were determined to ascertain the role of PER2 in DPC differentiation *in vitro*. After inducing odontogenic differentiation for 3 days, RT-qPCR showed that odontogenic differentiation markers, including *Dspp*, *Dmp1* and *Alp*, upregulated in OS group compared to the control group (Figs. 4B–4D). In OS-oe-PER2 group, *Dspp*, *Dmp1* and *Alp* dramatically elevated compared to OS-oe-NC group, indicating that overexpression of *Per2* resulted in significantly enhanced expression of differentiation markers of DPCs (Figs. 4B–4D). Similar results were obtained using western blotting. When protein levels were measured over seven days of odontogenic induction, DSPP and DMP1 went up significantly by 2-4 times in the OS group compared to Control group; the OS-oe-PER2 group revealed significantly higher levels of DSPP and DMP1 than the OS-oe-NC group, showing an approximately 2-fold increase (Figs. 4E–4G). Mineralized nodules were noted in OS group, but not in the Control group by Alizarin red staining (Fig. 4H). More mineralized nodules were observed in the OS-oe-PER2 group than in OS-oe-NC group, and semi-quantitative analysis also revealed that mineralized nodule formation was significantly enhanced in the OS-oe-PER2 group in comparison to OS-oe-NC group (Fig. 4H). The aforementioned results illustrate that overexpression of *Per2* promotes odontogenic differentiation of DPCs.

## Knockdown of *Per2* attenuates odontoblastic differentiation of DPCs *in vitro*

We knocked down *Per2* in DPCs and examined differentiation-related markers to further identify the contribution of PER2 to DPC differentiation (Fig. 5). Three different *Per2*-specific siRNAs (si-PER2-1,2,3) were transfected to knockdown *Per2*, as well as si-NC for negative control. Knockdown efficiency of *Per2* was measured *via* RT-qPCR and western blotting at 48 and 72-hours post-transfection, respectively (Fig. 5A). Knockdown efficiency of si-PER2-2 was the highest among the three si-RNAs, for both mRNA and protein

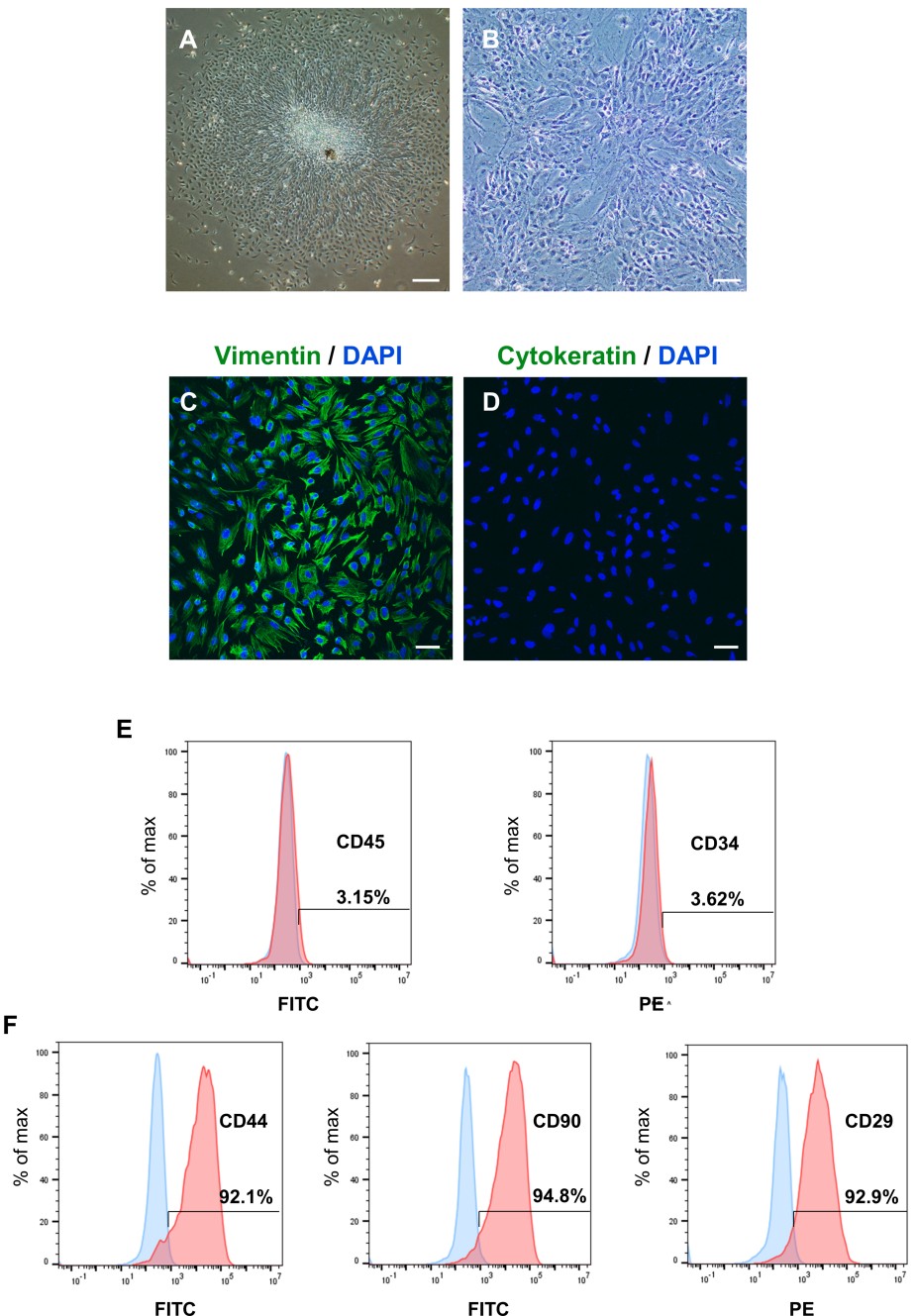

**Figure 2** **To isolate, culture and identify rat DPCs *in vitro*.** (A) After 3 days, primary DPCs migrated from pieces of dental papilla tissue. Scale bar: 200 μm; (B) After 2–3 passages, DPCs showed a homogeneous, large polygonal fibroblast-like appearance. Scale bar: 100 μm; (C) Immunofluorescence illustrated that DPCs positively expressed vimentin (green fluorescence), and the nuclei were stained blue by DAPI. Scale bar: 50 μm; (D) immunofluorescence staining showed that DPCs negatively expressed cytokeratin (green fluorescence), and the nuclei were stained blue by DAPI. Scale bar: 50 μm; (E–F) DPCs negatively expressed CD45 and CD34 and positively expressed CD44, CD90 and CD29; Blue: percentage of cells appearing in the specified fluorescent intensity range labelled with isotype control antibodies; red: percentage of cells appearing in the specified fluorescent intensity range labelled with markers antibodies.

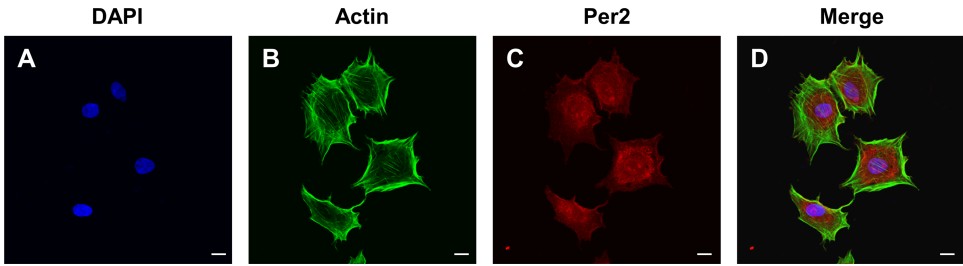

**Figure 3  Immunofluorescence staining reveals the expression level and subcellular localization of PER2 in rat DPCs *in vitro*.** (A) Nuclei are stained blue by DAPI; (B) Outlines of cells are stained green by Actin-Tracker; (C) PER2 immunofluorescence (red) visualized in both nuclei and cytoplasm of cells; (D) a merged figure. Scale bar: (A–D) 20 μm.

levels (>70% efficiency at the mRNA level, Fig. 5A). Therefore, si-PER2-2 was selected for subsequent assays and denoted as si-PER2 (Fig. 5).

After *Per2* was successfully knocked down, levels of differentiation-related markers were monitored to determine the role of PER2 on odontoblastic differentiation in DPCs *in vitro*. Three days after induction of odontogenic differentiation, RT-qPCR showed that the odontogenic differentiation markers (*Dspp*, *Dmp1* and *Alp*) dramatically upregulated in OS-si-NC group compared to Control-si-NC group (Figs. 5B–5D). Nevertheless, the aforementioned markers were significantly downregulated in the OS-si-PER2 group compared to the OS-si-NC group, with statistical significance (Figs. 5B–5D). Consistently, western blotting illustrated that DSPP and DMP1 considerably increased in the OS-si-NC group in comparison to the Control-si-NC group; nevertheless, a statistically significant decline in DSPP and DMP1 was observed in the OS-si-PER2 group compared to the OS-si-NC group (Figs. 5E–5G). Alizarin red staining indicated that mineralized nodules were present in the OS group after a 7-day-induced differentiation, whereas they were absent in the Control group (Fig. 5H). Reduced mineralized nodules were found in the OS-si-PER2 group compared with the OS-si-NC group, and semi-quantitative assays revealed a significant decrease in mineralized nodules in the OS-si-PER2 group compared to the OS-si-NC group (Fig. 5H). The aforementioned results indicated that *Per2* knockdown attenuated odontogenic differentiation of DPCs.

## Knockdown of *Per2* affects intracellular ATP content and ROS levels during DPCs differentiation *in vitro*

ROS levels and intracellular ATP content were examined in DPCs with *Per2* knockdown and odontogenic induction for 3/7 days to determine if the pro-differentiation effect of PER2 is associated with altered mitochondrial function (Fig. 6). The ROS levels of each group were determined using flow cytometry (Fig. 6A). Median fluorescence values for the Control-si-NC group are marked with a dashed line. Apparently, the median and mean fluorescence values of the OS-si-NC group is reduced compared to the Control-si-NC group, whereas the median and mean fluorescence of the OS-si-PER2 group is higher than that of the OS-si-NC group. As shown in Fig. 6B, ROS levels in the OS-si-NC group were reduced compared to the Control si-NC group; however, ROS levels in the OS-si-PER2 group were

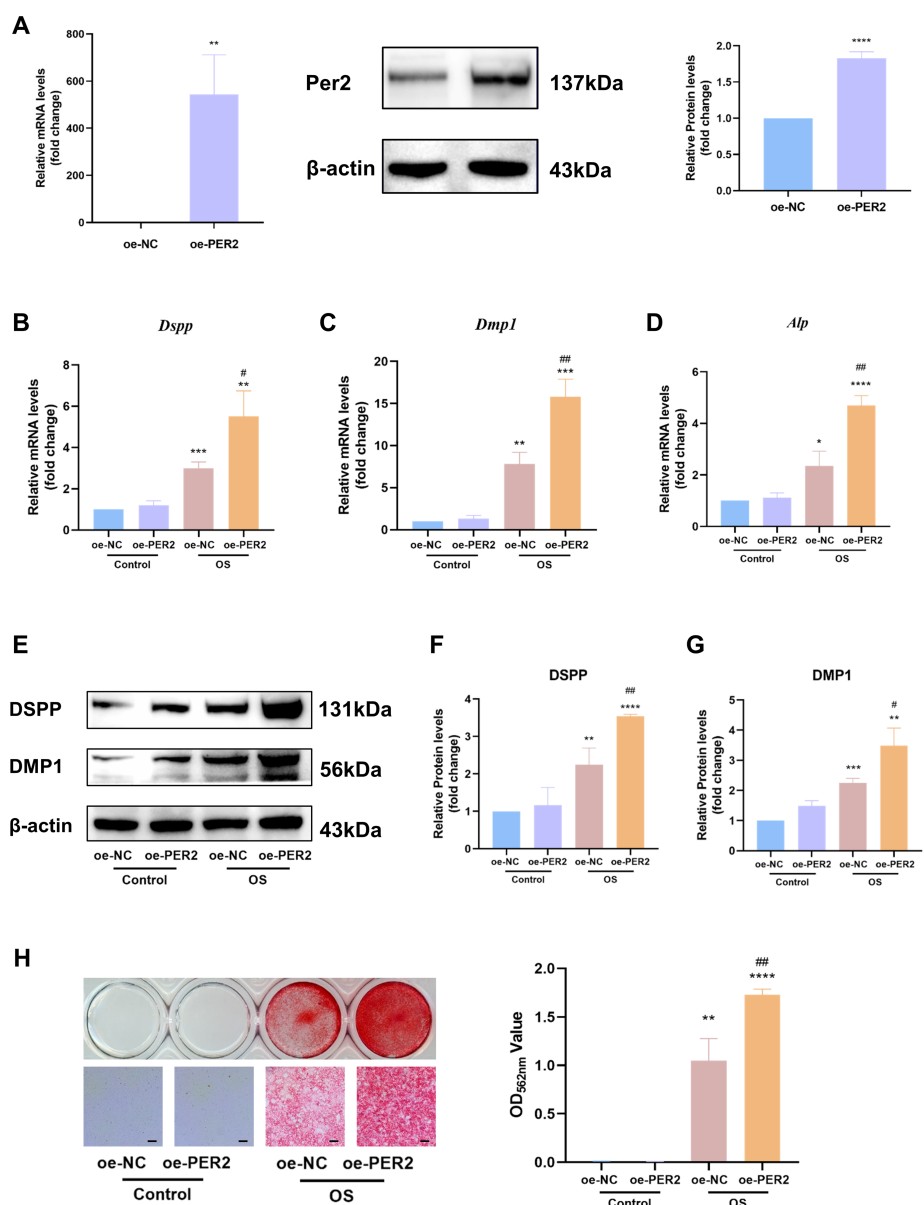

**Figure 4** **Overexpression of *Per2* enhances odontoblastic differentiation of DPCs *in vitro*.** PcDNA3.1-PER2 (oe-PER2) was transfected to overexpress *Per2*, as well as pcDNA3.1-NC (oe-NC) for negative control. Change the culture medium to conditioned medium (OS or basal medium as control) after 24 h. (A) Overexpression efficiency was determined through RT-qPCR and western blotting; (B–D) After 3 days of induction of differentiation, mRNA levels of *Alp, Dspp* and *Dmp1* were measured; Glyceraldehyde-3-phosphate dehydrogenase (GAPDH) served as the internal control; (E–G) after 7 days of induction of differentiation, proteins levels of DMP1 and DSPP, were measured; $\beta$-actin served as the internal control; (H) For 7 days of induction of differentiation, mineralized nodules were illustrated by alizarin red and semi-quantified by measuring absorbance at 562 nm after dissolution; Scale bar: 200 μm. Data are provided as mean ± SD ($n = 3$). Compared with oe-NC or Control-oe-NC groups, * $p < 0.05$, ** $p < 0.01$, *** $p < 0.001$, **** $p < 0.0001$. Compared with the OS-oe-NC group, # $p < 0.05$, ## $p < 0.01$, ### $p < 0.001$.

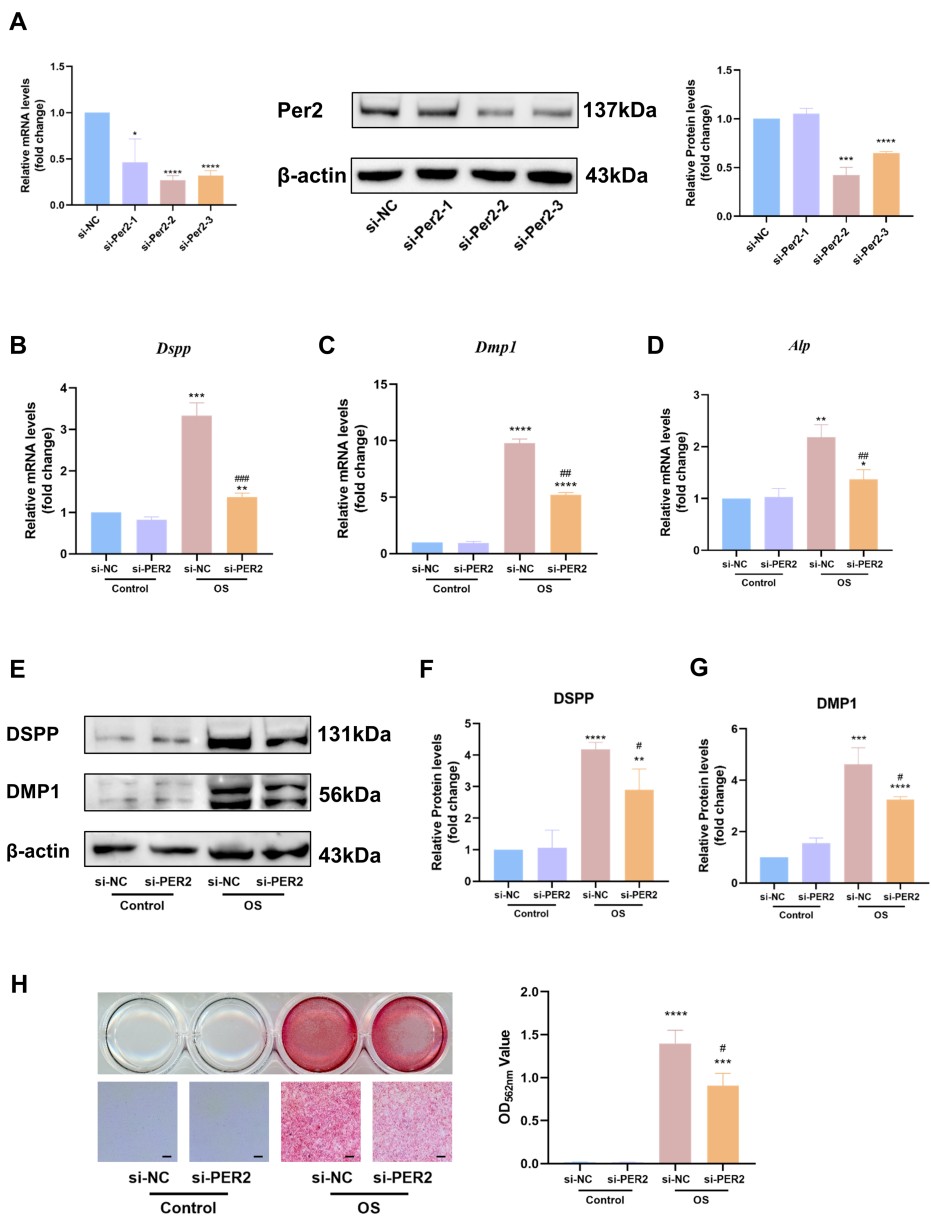

**Figure 5  Knockdown of *Per2* attenuates odontoblastic of DPCs *in vitro*.** Three different *Per2*-specific siRNAs (si-PER2-1,2,3) were transfected to knockdown *Per2*, as well as si-NC for negative control. Change the culture medium to conditioned medium (OS or basal medium as control) after 24 h. (A) Knockdown efficiency was evaluated through RT-qPCR and western blotting; (B–D) after 3 days of induction of differentiation, mRNA levels of *Alp, Dspp* and *Dmp1* were measured; GAPDH served as the internal control; (E–G) after 7 days of induction of differentiation, proteins levels of DMP1 and DSPP were measured; $\beta$-actin served as the internal control; (H) mineralized nodules were illustrated by alizarin red after a 7-day odontogenic induction and semi-quantified by measuring absorbance at 562 nm after dissolution; scale bar: 200 μm. Data are provided as mean ± SD ($n = 3$). In comparison to si-NC or Control-si-NC groups, * $p < 0.05$, ** $p < 0.01$, *** $p < 0.001$, **** $p < 0.0001$. In comparison to the OS-si-NC group, # $p < 0.05$, ## $p < 0.01$, ### $p < 0.001$.

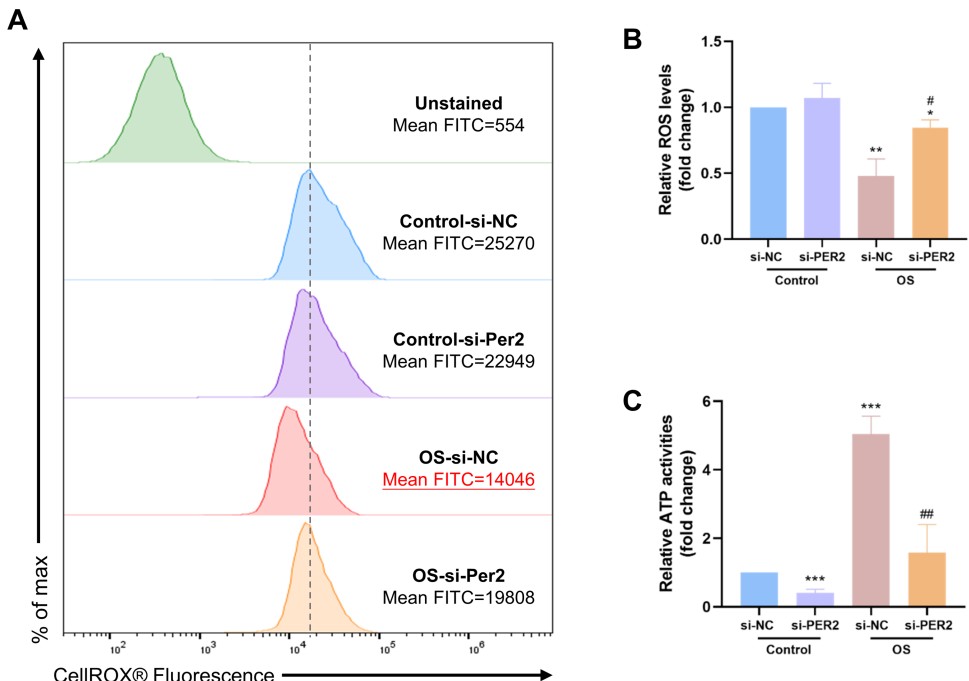

**Figure 6** **During DPCs differentiation *in vitro*, knockdown of *Per2* compromises mitochondrial respiratory function.** (A–B) After treatment of DPCs with knockdown of *Per2* and odontogenic induction for 3 days, ROS levels were measured using flow cytometry. The dotted line represents the position of the median of the Control-si-NC group. (C) After treatment of DPCs with *Per2* knockdown and odontogenic induction for 7 days, intracellular ATP levels were assessed and quantified for each group. Data are provided as mean ± SD ($n = 3$). In comparison to the Control-si-NC groups, * $p < 0.05$, ** $p < 0.01$, *** $p < 0.001$. In comparison to OS-si-NC group, # $p < 0.05$, ## $p < 0.01$.

elevated compared with the OS-si-NC group with statistical significance. These results suggested that ROS levels were downregulated in DPCs during differentiation. However, *Per2* knockdown may lead to an increase in ROS in DPCs during differentiation, which may affect differentiation. Intracellular ATP content of each group was also assessed (Fig. 6C). ATP content of the OS-si-NC group was visibly raised compared to the Control-si-NC group, whereas it was considerably declined in OS-si-PER2 group compared to OS-si-NC group (Fig. 6C). This suggests that intracellular ATP levels are upregulated in differentiated DPCs compared to undifferentiated DPCs, but knockdown of *Per2* may downregulate the ATP content of DPCs during differentiation, thereby suppressing differentiation.

## DISCUSSION

Genes in the *period* family (*Per*) encode circadian components (*Crosby & Partch, 2020*; *EmeklI et al., 2020*; *Miro et al., 2023*). There are three isoforms of the *per* family: *Per1-3*. *Per1* regulates oncogenesis by focusing on physiological rhythms, cell cycle, and DNA damage repair (*Gong, Tang & Yang, 2021*; *Wang et al., 2022*). Few studies have been conducted on *Per3*, which is primarily associated with sleep and circadian rhythm (*Archer et al., 2018*; *Li et al., 2023*). *Per2* is the most frequently studied member of the *Per* family. As a crucial and

powerful clock gene, *Per2* has been found to be involved in physio/pathological pathways in tissue development (*McQueen et al., 2018*), cell differentiation (*Huang et al., 2021*), cell metabolism (*Grimaldi et al., 2010*), cell aging (*Levine et al., 2020*), tumor development (*Guo et al., 2023*), oxidative stress (*Magnone et al., 2014*), and neurobiological activities (*Sayson et al., 2023*). For tooth development, PER2 has recently been identified as a potential regulator (*Huang et al., 2021*; *Jiang et al., 2022*; *Zheng et al., 2011*). This work offers a new perspective on the contribution of PER2 to DPC differentiation and dentin formation. We found a dramatic increase of PER2 expression as DPCs differentiate *in vivo*. Overexpression of *Per2* promoted DPC differentiation, while downregulation of *Per2* inhibited DPC differentiation. In addition, the pro-differentiation ability of PER2 in DPCs was associated with the intracellular ATP content and ROS levels.

Rat incisors grow throughout their life, and the entire process of tooth development can be observed in the incisors. The different stages of DPC differentiation can be visualized in a single section by immunohistochemistry of rat incisors. There is no continuous growth of the molars, and the cusps of the molars are irregular. It is difficult to obtain a single section of a molar that shows the progression of DPC from an undifferentiated to a differentiated state. Therefore, in this study, we chose the incisors for immunohistochemical assays, which showed that PER2 gradually upregulated along with DPCs differentiating into odontoblasts *in vivo*. PER2 was expressed to a small extent in undifferentiated DPCs but was highly expressed in differentiated mature odontoblasts. This expression pattern suggests that PER2 is associated with DPC differentiation. It is similar to a previous study showing that PER2 is strongly expressed in odontoblasts at different time points during tooth development, as demonstrated by immunohistochemical assays (*Zheng et al., 2011*).

*Per2* is a clock gene expressed in a 24-hour oscillatory pattern (*Jiang et al., 2022*). To minimize errors, samples for the cellular experiments were collected and assayed at the same time points by the same experimenter. The key finding of our study is that *Per2* overexpression promotes DPC differentiation under odontogenic induction, including elevated expression of odontoblastic biomarkers (DSPP, DMP1, ALP) and stronger mineralized nodule generation, while knockdown of *Per2* leads to the opposite result. Based on the bidirectional effects of overexpressing and knocking down *Per2*, it can be demonstrated that PER2 positively regulates odontogenic differentiation of DPCs. Furthermore, in groups without overexpression or knockdown of *Per2* (oe-NC group or si-NC group), odontogenic induction resulted in upregulation of DSPP, DMP1, and ALP expression and increased mineralized nodule formation, suggesting successful odontogenic differentiation of DPCs, independent of plasmid transfection or si-RNA transfection. Our findings show that PER2 modulates DPC differentiation and participates in tooth development. Similar to our findings, several papers have pointed out the involvement of PER2 in tooth development and dental hard tissue formation. Previous studies indicated that PER2 promotes ameloblast differentiation and enamel development (*Huang et al., 2021*). In addition, PER2 was found to be oscillatorily expressed in undifferentiated and differentiated DPCs over a 24-hour period (*Jiang et al., 2022*). PER2 is also associated with orthodontic tooth movement (*Hilbert et al., 2019*), and enamel defects caused by dental fluorosis (*Zou et al., 2022*).

Moreover, *Per2* knockdown leads to the downregulation of intracellular ATP content during differentiation of DPCs, which suggests that inhibition of DPC differentiation by *Per2* knockdown is linked to downregulation of ATP content and impaired energy metabolism in DPCs. The shift in energy metabolism brought about by mitochondrial activation is the key to MSC differentiation (*Chakrabarty & Chandel, 2021*; *Li et al., 2017*). The metabolic conditions of MSCs differ from those of differentiated cells; MSCs are primarily glycolysis-dependent, while differentiated cells are oxidative-phosphorylation (OXPHOS) dependent (*Chakrabarty & Chandel, 2021*; *Li et al., 2017*). When cell differentiation is initiated, mitochondria are excited through unidentified pathways, with OXPHOS replacing glycolysis as the dominant ATP source. This metabolic shift is crucial for MSC differentiation, and the ATP content increases as MSC differentiation proceeds (*Chakrabarty & Chandel, 2021*; *Kasahara & Scorrano, 2014*). Similar results were observed during the differentiation of DPCs. Our previous study confirmed that intracellular ATP content was elevated in DPCs with odontogenic differentiation compared to that in undifferentiated DPCs (*Zhang et al., 2018*). Inhibiting ATP production using rotenone, an inhibitor of mitochondrial complex I, can suppress DPC differentiation, suggesting that ATP content and mitochondrial activity are critical for DPC differentiation (*Zhang et al., 2018*). Notably, this research indicates that PER2 regulates intracellular ATP content of DPCs, which is consistent with previous findings in other cells. PER2 has been found to modulate intracellular ATP content in erythrocytes (*Sun et al., 2017*), colonic epithelial cells (*Chen et al., 2022*), hepatocytes (*Chen et al., 2009*) and cardiomyocytes (*Weng et al., 2021*).

In addition, we showed that knockdown of *Per2* dramatically increased ROS levels in DPCs at 3 days of odontogenic induction. *Per2* knockdown likely leads to impaired antioxidant capacity during DPC differentiation, making it difficult to remove excess ROS, resulting in blocked differentiation. Mitochondria dominate ROS production in mammalian cells. Similar to ATP, ROS act as signaling molecules for the mitochondria to modulate stem cell fate. During the early stages of bone marrow mesenchymal stromal cells (BMSCs) osteogenic differentiation (2–7 days), ROS levels were significantly downregulated to 25–50% of those in undifferentiated cells, and excess ROS inhibited BMSC osteogenic differentiation (*Chen et al., 2008*). Similar changes were observed in DPC differentiation. Our previous study found that ROS levels declined considerably in differentiated DPCs compared to control DPCs (*Zhang et al., 2018*). Furthermore, we found that PER2 is a negative regulator of ROS in DPCs, which is similar to the results of previous studies. For example, PER2 was reported to negatively regulate ROS levels in cardiomyocytes, spleens of elderly mice and ameloblasts (*He et al., 2022*; *Weng et al., 2021*; *Zou et al., 2022*).

## CONCLUSIONS

In summary, this study confirmed that PER2 expression levels in DPCs were continuously upregulated during odontoblastic differentiation *in vivo*. Incorporating overexpression and knockdown *Per2* assays, we demonstrated that PER2 positively modulated odontoblastic differentiation of DPCs *in vitro*. Furthermore, PER2 may affect both ATP content and

ROS levels while regulating DPC differentiation. The involvement of PER2 in DPC differentiation and tooth development may be related to the energy metabolism and antioxidant capacity of DPCs.

## ACKNOWLEDGEMENTS

The authors acknowledge the H&H research crew and faculty of the Guangdong Provincial Key Laboratory of Stomatology for their guidance on experimental techniques and instrumentation.

### Funding

This work was supported by the China Postdoctoral Science Foundation (Grant No. 2022M723593), the Guangdong Basic and Applied Basic Research Foundation (Grant No. 2022A1515110434), the National Natural Science Foundation of China No. 81870737, the Natural Science Foundation of Guangdong Province No. 2021A1515011779 and the Guangdong Financial Fund for High-Caliber Hospital Construction No. 174-2018-XMZC-0001-03-0125/D-02. The funders had no role in study design, data collection and analysis, decision to publish, or preparation of the manuscript.

### Grant Disclosures

The following grant information was disclosed by the authors:
China Postdoctoral Science Foundation: 2022M723593.
Guangdong Basic and Applied Basic Research Foundation: 2022A1515110434.
National Natural Science Foundation of China: 81870737.
Natural Science Foundation of Guangdong Province: 2021A1515011779.
Guangdong Financial Fund for High-Caliber Hospital Construction: 174-2018-XMZC-0001-03-0125/D-02.

### Competing Interests

The authors declare there are no competing interests.

### Author Contributions

- Haozhen Ma conceived and designed the experiments, performed the experiments, analyzed the data, prepared figures and/or tables, authored or reviewed drafts of the article, and approved the final draft.
- Xinyue Sheng performed the experiments, prepared figures and/or tables, and approved the final draft.
- Wanting Chen performed the experiments, prepared figures and/or tables, and approved the final draft.
- Hongwen He conceived and designed the experiments, authored or reviewed drafts of the article, and approved the final draft.
- Jiawei Liu performed the experiments, prepared figures and/or tables, and approved the final draft.

- Yifan He conceived and designed the experiments, authored or reviewed drafts of the article, and approved the final draft.
- Fang Huang conceived and designed the experiments, authored or reviewed drafts of the article, and approved the final draft.

## Animal Ethics

The following information was supplied relating to ethical approvals (*i.e.*, approving body and any reference numbers):

The present study was approved by the Institutional Animal Care and Use Committee, Sun Yat-sen University, China (No. SYSU-IACUC-2020-000511) and the Ethical Review Committee, Hospital of Stomatology, Sun Yat-Sen University, China (ERC-2013-15). This study adhered to the Ethical Principles of Animal Experimentation.

## Data Availability

The raw data are available in the Supplementary Files.

## Supplemental Information

Supplemental information for this article can be found online at http://dx.doi.org/10.7717/peerj.16489#supplemental-information.

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
