# Peer review of "PER2 regulates odontoblastic differentiation of dental papilla cells in vitro via intracellular ATP content and reactive oxygen species levels"

_PeerJ, doi:10.7717/peerj.16489_

## Round 0.1 · original submission · Minor Revisions

Please respond to the reviewers' comments point by point.

**Language Note:** PeerJ staff have identified that the English language needs to be improved. When you prepare your next revision, please either (i) have a colleague who is proficient in English and familiar with the subject matter review your manuscript, or (ii) contact a professional editing service to review your manuscript. PeerJ can provide language editing services - you can contact us at copyediting@peerj.com for pricing (be sure to provide your manuscript number and title). – PeerJ Staff

Reviewer 1 ·

Basic reporting

Generally, this study has been designed well with clear logic, the writing is professional, and the data quality is publishable.

Experimental design

1. Please label “Vimentin” in Figure 2C for clarity.

2. Can the authors perform immunofluorescent imaging with co-staining individual differentiation markers of different stages (DPC, pOB, iOB, and mOB) and PER2 in Figure 1 to show the dynamic spatial expression pattern of PER2?

3. In Figure 3, the authors identified that PER2 is strongly localized in the nuclei in the isolated DPCs. However, in the images in Figure 1, it is hard to see the strong nuclear localization. Will immunofluorescent imaging help this in Figure 1?

4. Please specify the statistical comparison test used in Figures 4-6.

Validity of the findings

Dental papilla cells (DPCs) are vital for tooth development, crafting dentin and pulp. PER2, previously noted to oscillate in DPCs in vitro, is expressed in odontoblasts. Its impact on DPCs' odontoblastic differentiation is unknown. This study investigates PER2's role, revealing its significant elevation during DPCs' odontoblastic differentiation in vivo. Overexpression bolsters odontogenic markers (DSPP, DMP1, ALP) and mineralized nodule formation. Conversely, PER2 downregulation hinders DPCs' differentiation. PER2 modulation also affects intracellular ATP and ROS during differentiation. In sum, PER2 positively influences DPCs' odontogenic differentiation, possibly linked to ATP and ROS levels.

Reviewer 2 ·

Basic reporting

In this study, PER2 was confirmed to participate in the regulation of DPCs differentiation into odontoblasts in vivo and in vitro. In addition, PER2 affected both intracellular ATP content and ROS levels during DPCs differentiation, indicating that the role of PER2 in DPCs differentiation and tooth development may be related to energy metabolism and antioxidant capacity. The research is intriguing with proper experimental design and performed rigorously. The experimental results are presented in a solid manner with conclusions well stated, linked to original research question . In addition, the manuscript is clearly written in professional, unambiguous language. However, I still have some points or concerns need to be addressed by the authors further.

Experimental design

1. It is highly recommended to conduct semi-quantitative analysis of immunohistochemistry results. In addition, incisors rather than molars were selected for immunohistochemistry, the reasons need to be stated in the discussion.
2. The differentiation and mineralization of odontoblasts are continuous processes. In this study, ROS and mRNA levels of odontogenic differentiation markers were measured on the 3th day after mineralization induction. Subsequently, ATP and protein levels were measured at the 7-day time point. Please explain the reason to select these specific time points.
3. The study used siRNA to knock down PER2 and it was quite effective. Please present the relevant siRNAs sequence information (5' to 3').
4. Previous studies have reported that PER2 was expressed in a 24-hour oscillatory pattern in DPCs in vitro. I wonder if the authors chose the same time point for sample collection and detection in cell experiments? Please describe in details in the experimental method and discussion.

Validity of the findings

no comment

Additional comments

1. In P12, Line295-296 and P13, LINE 336: "RT-qPCR showed that the odontogenic differentiation markers, including DSPP, DMP1 and ALP." Gene names are required to format with capitalizing the first letter, lower casing the rest of letters and italicizing. Please carefully check whether the gene names in this article were written properly.
2. In Figure 4, the same group were represented by pcDNA3.1-NC and oe-NC, pcDNA3.1-PER2 and oe-PER2, respectively. It is recommended to replace "pcDNA3.1-NC" with "oe-NC", and "pcDNA3.1-PER2" with "oe-PER2" in Figure 4 (A). This consistency in labeling will make it easier for readers to understand the figure, and avoid confusion.
3. The western blotting results in Figure 4 (A) and Figure 5 (A) need to be supplemented with grayscale analysis charts. In addition, it is recommended to indicate the molecular weight of the target protein.
4. Moreover, I suggest that that a brief introduction of PER family including Per1 with latest research progress might be added to the discussion, and the effects of PER2 overexpression on ATP and ROS need be confirmed too.

---

## Round 0.2 · accepted · Accept

The authors have fully responded to the reviewers' comments, and the quality of the manuscript has been significantly improved.